# Siderophore-inspired chelator hijacks uranium from aqueous medium

Alexander S. Ivanov [1], Bernard F. Parker [2,3], Zhicheng Zhang[2], Briana Aguila [4], Qi Sun[4], Shengqian Ma [4], Santa Jansone-Popova[1], John Arnold[2,3], Richard T. Mayes[1], Sheng Dai [1], Vyacheslav S. Bryantsev [1], Linfeng Rao[2] & Ilja Popovs [1]

Over millennia, nature has evolved an ability to selectively recognize and sequester specific metal ions by employing a wide variety of supramolecular chelators. Iron-specific molecular carriers—siderophores—are noteworthy for their structural elegance, while exhibiting some of the strongest and most selective binding towards a specific metal ion. Development of simple uranyl ($UO_2^{2+}$) recognition motifs possessing siderophore-like selectivity, however, presents a challenge. Herein we report a comprehensive theoretical, crystallographic and spectroscopic studies on the $UO_2^{2+}$ binding with a non-toxic siderophore-inspired chelator, 2,6-bis[hydroxy(methyl)amino]-4-morpholino-1,3,5-triazine ($H_2BHT$). The optimal $pK_a$ values and structural preorganization endow $H_2BHT$ with one of the highest uranyl binding affinity and selectivity among molecular chelators. The results of small-molecule standards are validated by a proof-of-principle development of the $H_2BHT$-functionalized polymeric adsorbent material that affords high uranium uptake capacity even in the presence of competing vanadium (V) ions in aqueous medium.

[1] Oak Ridge National Laboratory, Oak Ridge, TN 37831, USA. [2] Lawrence Berkeley National Laboratory, Berkeley, CA 94720, USA. [3] University of California, Berkeley, CA 94720, USA. [4] University of South Florida, Tampa, FL 33620, USA. These authors contributed equally: Alexander S. Ivanov and Bernard F. Parker. Correspondence and requests for materials should be addressed to V.S.B. (email: bryantsevv@ornl.gov) or to L.R. (email: lrao@lbl.gov) or to I.P. (email: popovsi@ornl.gov)

Achieving a balance between responsible and conscientious environmental stewardship of available natural resources and the ever-increasing demand for raw materials required to sustain higher standards of living is the quintessential requirement of sustainable development. Unlike terrestrial sources of minerals (mines, salt lakes etc.), seawater is almost an inexhaustible resource of base elements such as Na and Cl. Unsurprisingly, less-abundant elements present in the seawater, such as Mg, K, Ca, continue to be mined from oceans on an industrial scale at a significantly lower cost to the environment than alternative land-mining operations[1]. Numerous strategies have been proposed to recover even less-abundant high value elements from seawater, such as copper (Cu), nickel (Ni), vanadium (V), and uranium (U), however, their implementation remains elusive[2]. Uranium extraction from seawater especially has attracted a significant amount of attention from researchers in recent years, because of its predominant use as a fuel to produce nuclear power[3]. Despite its low dissolved concentration of only ~3 mg per ton of seawater, an estimated total content of uranium amounts to 4 billion tons, over 1000 times more than is available in all known terrestrial sources[4]. This almost inexhaustible supply of accessible uranium could fuel nuclear power generation for millennia even with the demand significantly increasing over time[5].

The pace of the development and commercialization of new adsorbent materials capable of cost-efficient and selective recovery of uranium from seawater has been slow, despite numerous attempts based on layered inorganic materials[6,7], engineered proteins[8,9], chelating resins[10–12], and polymeric adsorbent materials[13–15] that are currently in the development. The most successful adsorbents based on amidoxime and imide-dioxime (H$_3$IDO) functional groups (Fig. 1a) have shown some of the highest uranium adsorption capacities in excess of 4 g per gram of adsorbent[13]. Functional scaffolds based on amidoximes and imide-dioxime have emerged as widely used molecular receptors for uranyl ion (UO$_2^{2+}$) binding[16–19], as these are the functional groups predominantly formed in hydroxylamine-treated nitrile-based adsorbents, and particularly in polyacrylonitrile adsorbents. We and others have previously described the performance of adsorbents based on hydrophilic surface-grafted polyacrylonitrile[4,14,15,20,21]. However, the selectivity of these and other structurally-related materials still suffers from a much higher affinity towards vanadium (V) ions present in the seawater over uranium (VI), therefore requiring use of expensive reagents under harsh conditions for adsorbent regeneration and limiting their real-life implementation[22]. The development of more selective adsorbents containing superior functionalities is still required to overcome this bottleneck, especially materials that take advantage of environment friendly and non-toxic functionalities[23].

Siderophores, a class of naturally occurring (Fig. 1b) nitrogen- and oxygen-based donor group containing chelating compounds for iron sequestration present in bacteria and fungi that exhibit some of the highest binding affinities towards the metal, have long been recognized as potential ligands for f-block elements[24–26]. Artificial siderophores show record affinities towards iron, however, only a handful of f-element complexes have been characterized[27]. There have been speculations that iron in its +3 oxidation state (Fe$^{3+}$) and uranyl ion (UO$_2^{2+}$) interact very similarly with hydroxamate-based ligands, and therefore could be used as a proxy for one another[28]. Inspired by this hypothesis, as well as the desire to increase the selectivity of the adsorbents towards uranium, especially over vanadium, we have selected non-toxic and easy to prepare 2,6-bis[hydroxy(methyl)amino]-4-morpholino-1,3,5-triazine (H$_2$BHT)[29,30]—an artificial siderophore—as a ligand of choice (Fig. 1c).

Herein we report a comprehensive computational and experimental study on uranium binding by H$_2$BHT in solution and in solid state, starting from small-molecule investigations and ending up with a developed polymeric adsorbent material. Quantum chemical calculations corroborated by potentiometric titrations establish exceptionally high uranyl binding affinity and selectivity of the H$_2$BHT, making it one of the most compelling and unique functional units for uranium extraction and remediation technologies to date. We provide synthetic details and evidence that H$_2$BHT-functionalized polymers have high uranium adsorption capacity even in the presence of competing vanadium ions, presenting a novel approach towards the rational design and synthesis of economically attractive tailor-made adsorbents for selective metal ion recognition and sequestration from aqueous media.

## Results

**Crystallographic and quantum chemical calculation studies**. In order to find the representative binding mode between H$_2$BHT chelator[31] and uranium, the 2,6-bis[hydroxy(methyl)amino]-4-morpholino-1,3,5-triazine ligand was synthesized and tested for complexation with uranyl (Supplementary Methods). Single crystals of UO$_2$(BHT) (Fig. 2a) were readily obtained from the stoichiometric aqueous solution of uranyl nitrate and H$_2$BHT on standing over the period of three weeks, however, all attempts to isolate crystals of UO$_2$(BHT)$_2^{2-}$ suitable for X-ray diffraction studies were unsuccessful[32]. As seen from Fig. 2a, H$_2$BHT coordinates to uranium in a tridentate fashion, similar to its coordination with other metals, however, the density functional theory (DFT) calculations at the M06/SSC/6-311 + + G** level of theory indicate that binding modes can vary significantly depending on protonation states of the ligand (Supplementary Fig. 1). In agreement with the crystallographic data (Supplementary Tables 1, 2), the DFT optimized UO$_2$(BHT) complex represents a neutral five-coordinate uranyl species (one H$_2$BHT and two water molecules in the first coordination sphere) with relatively short

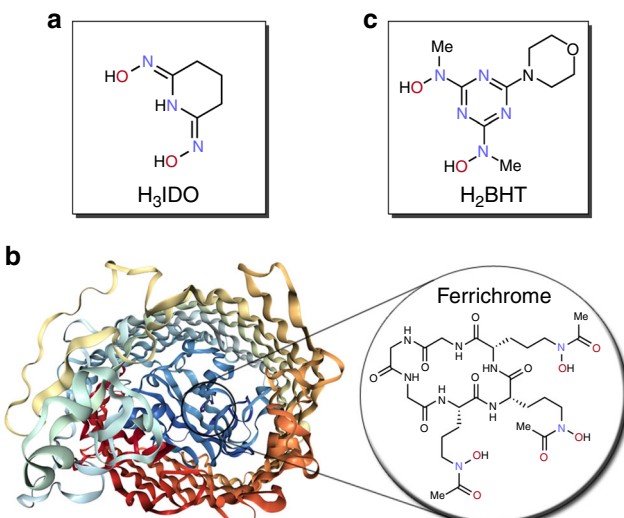

**Fig. 1** Structures of ligands for selective recognition of metal ions. **a** Glutaroimide-dioxime (H$_3$IDO) ligand, which is the major functional unit of polyamidoxime adsorbents for uranium recovery from seawater[22]. **b** Cartoon representation of the ferric hydroxamate uptake (FhuA) protein (image from the RCSB PDB (www.rcsb.org) of PDB ID 1BY5[46]) with the highlighted structure of ferrichrome—a natural siderophore. **c** Bis-(hydroxylamino)-1,3,5-triazine (H$_2$BHT)—an artificial siderophore investigated in this study

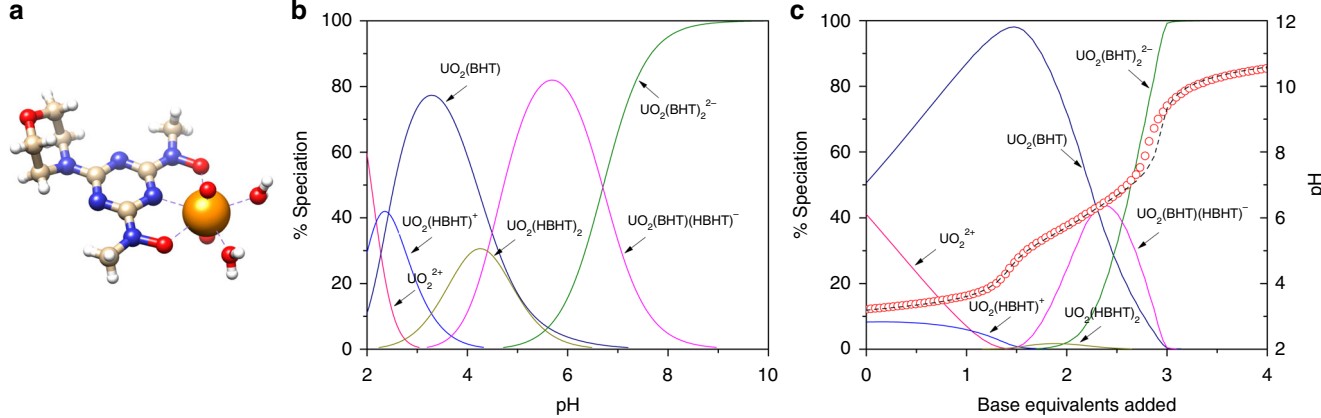

**Fig. 2** Structural and potentiometric studies of the H$_2$BHT complexation with uranyl. **a** Single crystal structure of UO$_2$(BHT) shown as a ball-stick model (U(VI), orange; O, red; N, navy blue; C, beige; H, white). **b** Calculated speciation as a function of pH; conditions: [U(VI)] = 0.2 mM, [H$_2$BHT] = 0.4 mM. **c** Potentiometric titration for the complexation of uranium with H$_2$BHT at 25 °C and $I$ = 0.5 M (NaCl); base equivalents are with respect to ligand; observed (circles) and calculated (−) pH (right axis) with corresponding speciation (left axis); conditions: [U(VI)] = 0.362 mM, [H$_2$BHT] = 0.786 mM

---

**Table 1 Theoretically calculated and experimental stability constants (log $\beta$) of uranyl complexes with H$_2$BHT and H$_3$IDO ligands**

| Aqueous Species, Reaction | log $\beta^{\text{theor a}}$ | log $\beta^{\text{expt b}}$ |
|---|---|---|
| Bis-(hydroxylamino)-1,3,5-triazine (H$_2$BHT) ligand | | |
| UO$_2^{2+}$ + BHT$^{2-}$ ⇌ UO$_2$(BHT) | 20.3 | 17.47 ± 0.27 |
| UO$_2^{2+}$ − H$^+$ + BHT$^{2-}$ ⇌ UO$_2$(BHT)(OH)$^-$ | n.a.$^d$ | 7 ± 1 (est.)$^e$ |
| UO$_2^{2+}$ + H$^+$ + BHT$^{2-}$ ⇌ UO$_2$(HBHT)$^+$ | 21.6 | 19.9 ± 2.5 |
| UO$_2^{2+}$ + 2BHT$^{2-}$ ⇌ UO$_2$(BHT)$_2^{2-}$ | 30.5 | 26.7 ± 0.56 |
| UO$_2^{2+}$ + H$^+$ + 2BHT$^{2-}$ ⇌ UO$_2$(HBHT)(BHT)$^-$ | 36.4 | 33.4 ± 0.50 |
| UO$_2^{2+}$ + 2 H$^+$ + 2BHT$^{2-}$ ⇌ UO$_2$(HBHT)$_2$ | 41.9 | 37.8 ± 2 |
| Glutaroimide-dioxime (H$_3$IDO) ligand$^c$ | | |
| UO$_2^{2+}$ + HIDO$^{2-}$ ⇌ UO$_2$(HIDO) | 19.9 | 17.8 ± 1.1 |
| UO$_2^{2+}$ + H$^+$ + HIDO$^{2-}$ ⇌ UO$_2$(H$_2$IDO)$^+$ | 25.3 | 22.7 ± 1.3 |
| UO$_2^{2+}$ + 2 HIDO$^{2-}$ ⇌ UO$_2$(HIDO)$_2^{2-}$ | 29.5 | 27.5 ± 2.3 |
| UO$_2^{2+}$ + H$^+$ + 2HIDO$^{2-}$ ⇌ UO$_2$(H$_2$IDO)(HIDO)$^-$ | 35.2 | 36.8 ± 2.1 |
| UO$_2^{2+}$ + 2H$^+$ + 2HIDO$^{2-}$ ⇌ UO$_2$(H$_2$IDO)$_2$ | 45.0 | 43.0 ± 1.1 |

$^a$Calculated at 25 °C and $I$ = 0
$^b$Obtained at 25 °C and $I$ = 0.5 M
$^c$Experimental (25 °C, $I$ = 0.5 M) and computational (25 °C, $I$ = 0) data are taken from our past works[35,38]
$^d$n.a.—no available value
$^e$est.—estimated value based on speciation modeling

---

U—O (2.35 Å) and U—N (2.42 Å) bond lengths between uranium and two deprotonated hydroxylamino oxygens and the central pyridine-like nitrogen atom, respectively. Subsequent protonation of this complex leads to the substantial elongation of one U—O (2.63 Å) bond with the protonated hydroxylamine group and slight shortening of the other U—O (2.21 Å), while U—N (2.46 Å) distance remains almost unchanged, indicating strong U—N interactions due to the high electron density on the triazine nitrogen as a result of resonant contribution of electrons from exocyclic nitrogen atoms in the ring[33]. Interactions of UO$_2^{2+}$ with two H$_2$BHT chelators lead to formation of the corresponding 1:2 complexes depicted in Supplementary Fig. 1, which is in agreement with the experimental data[32].

Having established the most stable structures of the uranyl complexes in solution, we proceeded to calculate the key thermodynamic parameters, which then were used in our computational protocol[34,35] for predicting stability constants (log $\beta^{\text{theor}}$). This procedure enables us to carry out in silico prediction of log $\beta$ for uranyl complexes with the H$_2$BHT ligand and compare the obtained values with those of uranyl systems with the imide-dioxime (H$_3$IDO) functionality, which is reputedly responsible for the extraction of uranium from

seawater using the current generation of amidoxime-derived sorbents[4]. The computational results (log $\beta^{\text{theor}}$) in Table 1 indicate very similar binding strengths of H$_2$BHT and H$_3$IDO. On one hand, the electron-withdrawing effect of the aromatic triazine ring in H$_2$BHT lowers the electron density and so the basicity of its hydroxylamine groups, which would result in a weaker complexation ability of H$_2$BHT with UO$_2^{2+}$ than that of H$_3$IDO. On the other hand, this electron-withdrawing effect can be offset by the presence of electron-donating methyl groups at the nitrogen atoms in H$_2$BHT, explaining comparable uranyl binding affinities of the triazine hydroxylamine and imide-dioxime ligands. Natural bond orbital (NBO) analysis further confirms strong uranyl binding by both functionalities, revealing effective metal–ligand interactions via dative σ-bonds for the neutral 1:1 H$_2$BHT and H$_3$IDO uranyl complexes (Supplementary Table 3). The protonated UO$_2$(HBHT)$^+$, UO$_2$(HBHT)(BHT)$^-$, and UO$_2$(HBHT)$_2$ complexes are generally 3–4 orders of magnitude weaker than the respective uranyl complexes formed with imide-dioxime. This trend can be rationalized by the structural differences between H$_2$BHT and H$_3$IDO (Fig. 1a, c). While in the imide-dioxime complexes the protons of both oxime groups can undergo transfer from

the oxygen to nitrogen atom to increase the binding strength with uranyl, a similar tautomeric rearrangement in $H_2BHT$ is prevented by methyl groups.

**Potentiometric titrations and NMR spectroscopic studies.** The $H_2BHT$ ligand protonation constants ($pK_a$) were determined in a 0.5 M NaCl ionic medium (Supplementary Table 4), and are in excellent agreement with previously reported values in other media[33]. It is worth noting that $H_2BHT$ is more acidic than imide-dioxime, with $pK_{a1} = 8.0 \pm 0.3$, indicating that in oceanic environment with pH 8.0–8.3 approximately half the ligand is already in a mono-deprotonated state that can facilitate metal ion complexation[23,36,37]. Potentiometric titrations were performed to verify the presence of theoretically predicted complexes in the uranyl-$H_2BHT$ aqueous systems and the values of their respective equilibrium constants $\log \beta^{expt}$. The modeled speciation diagram at a 1:2 ratio as a function of pH is shown in Fig. 2b and a representative titration curve is provided in Fig. 2c, while the stability constants are summarized in Table 1. Our results in Fig. 2b indicate that both 1:1 and 1:2 U:$H_2BHT$ species are formed, dependent on pH of the solution. The same stoichiometries and charges for uranyl complexes are observed with both imide-dioxime ($H_3IDO$) and $H_2BHT$ making comparison between the corresponding species straightforward. Despite the overall stability constants being slightly higher for $H_3IDO$ complexes, the more acidic nature (lower first $pK_{a1}$ value) of $H_2BHT$ indicates that both ligands have similar binding affinity towards uranyl when starting from the ligand protonation states that are dominant at pH 7–9. Moreover, $H_2BHT$ is much more selective for uranium (VI) over vanadium (V) compared to the $H_3IDO$ ligand, which is known to form strong 1:2 non-oxido vanadium complex $V(IDO)_2^-$ with very high stability constant value of 53.0[22,38]. In contrast, $H_2BHT$ can only form a 1:1 dioxovanadium complex $(VO_2(BHT)-)$[33], which is significantly weaker ($\log \beta =$ 17.9) than the 1:2 uranyl-$H_2BHT$ complexes (Table 1). The inability of $H_2BHT$ to form a hypothetical 1:2 non-oxido vanadium complex $V(BHT)_2^+$ was further supported by DFT calculations (Supplementary Note 1 and Supplementary Table 5). The simulated speciation diagrams in Supplementary Fig. 2 indicate that there is no $V(BHT)_2^+$ complex formation over the entire pH range even in the presence of a large excess of the ligand. It is worth noting that in addition to vanadium, iron ions ($Fe^{3+}$) form strong complexes with imide-dioxime, exhibiting high stability constants, e.g., $\log \beta$ ($Fe(HIDO)_2^-$) = 36.0[39], which is 8.5 orders of magnitude higher than $\log \beta$ for the corresponding $UO_2(HIDO)_2^{2-}$ complex (Table 1). A comparison of formation constants for the 1:2 complexes with bis-(hydroxylamino)-1,3,5-triazine functionality ($\log \beta$ ($Fe(BHT)_2^-$) = 25.3[29] and $\log \beta$ ($UO_2(BHT)_2^{2-}$ = 26.7) revealed comparable binding affinity of $H_2BHT$ toward $Fe^{3+}$ and $UO_2^{2+}$, suggesting possible competition of iron ions with uranyl for adsorption to the $H_2BHT$-functionalized sorbent material. Nevertheless, it is reasonable to assume that potentially adsorbed iron species would not strongly affect the recyclability of $H_2BHT$-based polymer, since iron complexes could be efficiently hydrolyzed and stripped from the polyamidoxime fibers under acidic pH conditions without damaging the adsorbent[40].

To shed an additional light on the solution-state uranyl-ligand complexation behavior, NMR experiments were performed to corroborate the results of potentiometric titrations and computations pertaining to solution speciation in the $H_2BHT$-uranyl system (Supplementary Fig. 3). In aqueous solution, the $UO_2(BHT)$ complex is the predominant species formed under acidic to neutral conditions (Fig. 2b). However, once crystallized from solution as $UO_2(BHT)$ (crystal structure), the complex

becomes insoluble in neutral or acidic water, indicative of a neutral, low-polarity species. Surprisingly, upon addition of base (3 equivalents, final pH = 12) to the $UO_2(BHT)$ complex in water, it re-dissolves. We propose that the complex is deprotonated in solution forming the anionic complex $UO_2(BHT)(OH)^-$ (Table 1). This monoanionic species is not present in speciation models from the potentiometric titrations. However, the NMR experiments were only performed with a 1:2 or larger uranyl to ligand ratio. When a formation constant of log $\beta^{expt} = 7.0$ for $UO_2(BHT)(OH)^-$ is added to the model, the complex is not observed at 1:2 or higher uranyl to ligand ratios, however, at a 1:1 ratio $UO_2(BHT)(OH)^-$ appears at pH = 9 and is the only species present in the solution at pH = 12. Further addition of the ligand results in the solution becoming paler in color, and the NMR spectra confirm the formation of new species, that we assign as $UO_2(BHT)_2^{2-}$, with excess amount of free ligand and very small amounts of the $UO_2(BHT)$ complex still present. The $^{13}C$ spectra is consistent with the formation of $UO_2(BHT)_2^{2-}$ species in solution (Supplementary Fig. 4), with 2D $^1H$-$^{13}C$ HSQC spectrum further corroborating this assignment (Supplementary Fig. 5). The results of potentiometric titrations and NMR spectroscopy were further supplemented by UV-Visible absorption studies (Supplementary Fig. 6 and Supplementary Note 2), providing additional confirmation of the observed speciation in solution over a wide pH range.

**Adsorbent material synthesis and uranium sorption studies.** As with other hydroxylamine-derived ligands, the direct incorporation of such species via free radical or cationic polymerization is not feasible due to radical quenching and strong nucleophilicity of the hydroxylamine group, the presence of relatively acidic protons precludes the use of anionic polymerization. Therefore, an alternative approach based on a stepwise construction of $H_2BHT$ core via post-synthetic functionalization has been developed instead[41]. The synthesis of $H_2BHT$-decorated polymeric adsorbent was commenced with the polyethylene-graft-polyacrylic acid hollow gear fibers prepared employing our previously developed radiation induced graft polymerization (RIGP) technique (Supplementary Fig. 7)[42]. The activation of carboxylic group with thionyl chloride under mild conditions provided acyl chloride-functionalized polymeric material. Subsequent coupling reaction with mono N-tert-butyl carbonyl (BOC) protected piperazine derivative resulted in the formation of the monoamide-monocarbamate intermediate. Next, the deprotection of the N-BOC substituted amine by treating with excess trifluoroacetic acid resulted in the formation of the corresponding ammonium trifluoroacetate, which upon addition of excess amounts of N,N-diisopropylethylamine (DIEA) and 1,3,5-trichlorotriazine afforded the aminodichlorotriazine advanced intermediate[43]. The final reaction of the polymer-bound aminodichlorotriazine with an excess amount of N-methyl hydroxylamine free base in tetrahydrofuran (prepared in situ by treating corresponding hydrochloride salt with DIEA), provided the $H_2BHT$-functionalized polymeric adsorbent in five linear steps, constituting the first proof-of-principle synthesis of an artificial siderophore-embedded uranium adsorbent. The scanning electron microscopy (SEM) images reveal similar polymer morphologies for the pristine polyethylene-graft-polyacrylic acid (Fig. 3a) and the $H_2BHT$-modified polymer (Fig. 3b), indicating that the initial structure of the fiber was not damaged during the synthesis process. In addition, the structure of the $H_2BHT$-polymer was confirmed by Fourier-transform infrared spectroscopy (FT-IR) (Supplementary Fig. 8), the incorporation of $H_2BHT$ ligand into the polymer structure was further corroborated by $^{13}C$ cross polarization/magic-angle spinning (CP/MAS)

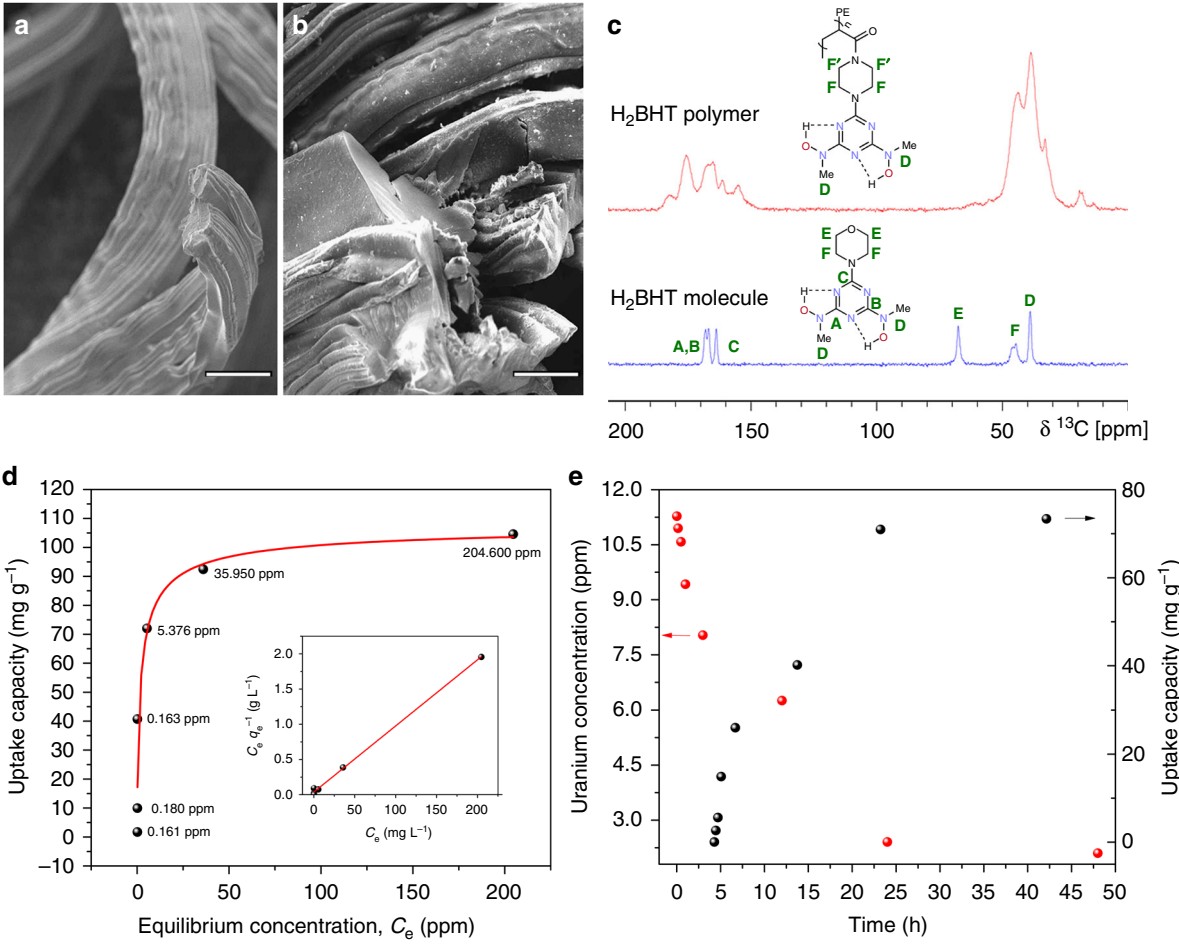

**Fig. 3** Characterization and performance of the H$_2$BHT polymeric adsorbent. **a**, **b** SEM images for the starting material (polyethylene-graft-polyacrylic acid) and the H$_2$BHT adsorbent, respectively. Scale bars, 50 μm. **c** δ $^{13}$C CP/MAS solid-state NMR spectra of H$_2$BHT polymer and H$_2$BHT ligand spun at 15 kHz. **d** Uranium sorption isotherm (inset displays the linear regression by fitting the equilibrium data with the linear form of the Langmuir adsorption model). All the fits (red lines) have R$^2$ values higher than 0.98. The equilibrium concentration (ppm) of uranyl ions is given for each data point. **e** Kinetics of uranium removal from aqueous solution with an initial concentration (10 ppm, 400 mL) at pH ~6, and adsorbent material (5 mg)

NMR spectroscopy (Supplementary Fig. 9). The overlay of spectra corresponding to the H$_2$BHT molecule and the polymer, depicted in Fig. 3c, clearly shows the presence of the three triazine carbon atoms (A, B, and C) in the polymer. The signals from piperazine carbons F and F′, as well as the signals from $N$-methyl carbons D overlap with signals from the carbons present in the polyethylene and polyacrylamide backbone, falling in the same 10–60 ppm region, consistent with solution-state NMR of the ligand[29].

With the H$_2$BHT polymer adsorbent in hand, we investigated the uranium (VI) binding by this material using a variety of spectroscopic techniques, including X-ray photoelectron spectroscopy (XPS) and elemental distribution mapping via energy-dispersive X-ray (EDX) spectroscopy analysis. Uranium inclusion within the adsorbent is evident from the characteristic U 4f signals in the XPS spectra of the uranyl reacted polymer sample (Supplementary Fig. 10). The EDX elemental mapping results (Supplementary Fig. 11) confirm the existence of uranium species on the surface of the H$_2$BHT functionalized polymer with a homogeneous distribution of the extracted uranium throughout the material. Furthermore, FT-IR spectrum of the uranyl contacted sample (Supplementary Fig. 12) exhibits a significant red-shifted peak that corresponds to the antisymmetric stretch vibrational mode of $[O=U=O]^{2+}$, indicating strong interactions between uranyl and the H$_2$BHT functional groups in the developed polymeric adsorbent.

Next we tested uranyl ion extraction ability from aqueous solutions by varying metal concentrations, to shed light on the affinity of this material towards uranium (VI) species. The adsorption isotherm (pH of the solution was optimized at 6, Supplementary Fig. 13) was obtained to assess the overall capacity of the adsorbent by allowing it to equilibrate with solutions containing varying $UO_2^{2+}$ concentrations ranging from 1 to 250 ppm and a phase ratio of 0.25 mg mL$^{-1}$. The resulting uranium adsorption isotherm in Fig. 3d is in excellent agreement with the Langmuir model, exhibiting a correlation coefficient $R^2 = 0.988$. As was envisioned from small-molecule studies, the H$_2$BHT adsorbent material has high affinity towards uranyl ion in aqueous solution with the maximum uptake capacity of 105 mg g$^{-1}$ with an equilibrium concentration of 205 ppm (experimental data used to plot the uranium sorption isotherm are given in Supplementary Table 6). To probe the uranium sorption kinetics of H$_2$BHT, the aliquots were withdrawn from a 10 ppm solution of uranyl at the appropriate time intervals and analyzed by Inductively Coupled Plasma-Optical Emission Spectroscopy (ICP-OES) (Supplementary Methods). The uranium uptake by the H$_2$BHT polymer manifested by the decrease of metal concentration still present in the solution. As seen from Fig. 3e, the initial concentration was rapidly lowered and equilibrium capacity of the polymeric adsorbent was reached within 24 h, signifying relatively fast uranium uptake kinetics. In addition, a competitive

adsorption test was performed to probe the selectivity profile of the developed material towards uranium (VI) versus vanadium (V) species in solution. The adsorbent sample (6.5 mg) was stirred overnight in 200 mL solution (pH 6) with equal concentrations (10 ppm) of uranium ($UO_2(NO_3)_2.6H_2O$) and vanadium ($Na_3VO_4$). Impressively, the uranium concentration dropped significantly from 10 ppm to ~0.15 ppm despite the high concentration of competing vanadium (V) species in solution (Supplementary Tables 7-9 and Supplementary Note 3). From these results, we conclude that the $H_2BHT$ polymeric chelator has superior binding characteristics for uranium (VI) over vanadium (V), affording selective recognition of uranyl in the mixture of metal ions. In contrast to the currently used state-of-the-art polyamidoxime adsorbents[44], which possess very low recyclability due to highly acidic conditions required during the elution process[40], the $H_2BHT$-based adsorbent can be completely regenerated by aqueous carbonate treatment (1 M solution of $Na_2CO_3$) without damaging the material (Supplementary Table 7 and Supplementary Note 3).

## Discussion

We have performed combined comprehensive studies on the synthesized and characterized uranyl-selective adsorbent based on $H_2BHT$ artificial siderophore ligand. Simple synthetic procedure in combination with the use of cheap and commercially available starting materials, as well as efficient post-synthetic functionalization strategy resulted in an efficient adsorbent material. Solid-state NMR and IR spectroscopic evidence point out to the presence of $H_2BHT$ ligands on the polymer. The material exhibits preferential absorption of uranium over vanadium in the competition experiment, with the uranyl capacity in excess of 100 mg per gram of adsorbent without apparent inhibition by the vanadium (V) ions. Ab initio calculations complemented by potentiometric titration and NMR experiments provided strong rational for the observed uranyl capacity and selectivity. The $H_2BHT$ chelator shows much stronger affinity towards uranium (VI) over vanadium (V), compared to the currently used tridentate O,N,O-donors like imide-dioxime ($H_3IDO$). Importantly, unlike $H_3IDO$-based adsorbents, the $H_2BHT$-derived material does not form strong and persistent non-oxido vanadium (V) complexes, instead remaining as weaker monomeric or bridging dioxo complexes[33,38,45], while exhibiting almost identical affinity towards uranyl ions. As a result, the adsorbed uranium could be easily recovered from the polymer using simple aqueous carbonate treatment, while the adsorbent material can be readily regenerated by aqueous base treatment and further reused.

In addition to the successful synthesis and testing of the adsorbent, complete agreement between computational and experimental results from small-molecule studies have been validated by a proof-of-principle synthesis of the adsorbent material that exhibits the same features as small-molecule ligand. This study ushers in a practical approach towards the polymer design and synthesis of adsorbent materials decorated with the tailor-made ligands. Current efforts on extending this strategy to applications in water remediation and sequestration of other valuable metals are ongoing and will undoubtedly result in even deeper understanding of the underlying factors affecting the selectivity and efficiency of metal sequestration from complex aqueous media.

## Methods

**Materials and measurements**. Commercially available reagents were purchased and were used without additional purification. $^1H$ NMR spectra were recorded on a Bruker Avance-400 (400 MHz) spectrometer. Chemical shifts are expressed in ppm downfield from TMS at $\delta = 0$ ppm, and J values are given in Hz. $^{13}C$ (100.5 MHz) cross-polarization magic-angle spinning (CP-MAS) NMR experiments were recorded on a Bruker Avance-400 (400 MHz) spectrometer equipped with a magic-angle spin probe in a 4-mm $ZrO_2$ rotor. Scanning electron microscopy (SEM) was performed on JEOL SM-6060 Scanning Electron Microscope. IR spectra were recorded on a Perkin-Elmer Frontier FT-IR spectrometer. ICP-OES was performed on a Perkin-Elmer Elan DRC II Quadrupole instrument. EDX mapping were performed on a Hitachi SU 8000. XPS spectra were obtained on a Thermo ESCALAB 250 with Al Kα irradiation at $\theta = 90°$ for X-ray sources, and the binding energies were calibrated using the C1s peak at 284.9 eV. Details on the synthesis, potentiometric titrations, UV-Vis, NMR, X-ray crystallography, and computational studies are given in the Supplementary Methods.

## Data availability

The X-ray crystallographic coordinates for $UO_2(BHT)$ structure reported in this study have been deposited at the Cambridge Crystallographic Data Centre (CCDC), under deposition number 1854619. These data can be obtained free of charge from The Cambridge Crystallographic Data Centre via www.ccdc.cam.ac.uk/data_request/cif. All data supporting the findings discussed here are available within the paper and its Supplementary Information files, or from the corresponding authors upon request. Additional data can be provided by the authors upon reasonable request.

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

## Acknowledgements

This work was sponsored by the U.S. Department of Energy, Office of Nuclear Energy, under Contract DE-AC05-00OR22725 with Oak Ridge National Laboratory, managed by UT-Battelle, LLC, and Contract DE-AC02-05CH11231 with Lawrence Berkeley National Laboratory. The computational work (A.S.I. and V.S.B.) used resources of the National Energy Research Scientific Computing Center and the Compute and Data Environment for Science (CADES) at the Oak Ridge National Laboratory, both of which are supported by the Office of Science of the U.S. Department of Energy under contracts DE-AC02-05CH11231 and DE-AC05-00OR22725, respectively. The synthesis and characterization of the polymeric adsorbent (S.J.-P., I.P., S.D., and R.T.M.), thermodynamic and spectroscopic experimental work on small-molecule standard (B.F.P., Z.Z., J.A., and L.R.) and all computational studies (A.S.I. and V.S.B.) were supported by the Fuel Cycle Research and Development Campaign (FCRD)/Fuel Resources Program, Office of Nuclear Energy, U.S. Department of Energy (USDOE). Experimental work of S.J.-P. was supported by the U.S. Department of Energy, Office of Science, Basic Energy Sciences, Chemical Sciences, Geosciences, and Biosciences Division (ERKCC08). Experimental work by B.A., Q.S., and S.M. was supported by the DOE Office of Nuclear Energy's Nuclear Energy University Program (Grant No. DE-NE0008281). B.A. and S.M. would like to thank Dr. Eric J. Werner at University of Tampa for use of the ICP-OES instrument.

## Author contributions

A.S.I. and B.F.P. contributed equally. A.S.I. and V.S.B. performed the computations. B.F.P., Z.Z. and L.R. conducted the potentiometric and spectroscopic studies on the small-molecule standard. L.R. supervised the work by B.F.P. and Z.Z. at LBNL. J.A. supervised the research of B.F.P. at University of California, Berkeley, and at LBNL. I.P. and S.J.-P. performed synthesis and characterization of the adsorbent with input from S.D. and R.T.M. B.A., Q.S. and S.M. contributed to uranium uptake capacity, selectivity, XPS, and EDX studies of the adsorbent. I.P. and A.S.I. conceptualized the project. I.P., A.S.I. and B.F.P. wrote the original draft. The final manuscript was written through contributions from all authors. All authors discussed the results and commented on the manuscript.
