## [Peer Review File · Nature Communications]

Reviewers' comments:

Reviewer #1 (Remarks to the Author):

The reviewer finds the current manuscript (ms) has many deficiencies that need to be addressed. Although the authors highlight the first preparation of uranyl-selective adsorbent based on H2BHT artificial siderophore ligand, the characterization and the sorption study are fairly routine and oversimplified. Specially,

1. How about the nature of the surface modification? The current characterizations can not support that H2BHT ligand was modified by chemical binding?

2. The details on the sorption experiments were not fully provided. The solution pH, for example, was not indicated in sorption isotherm and selectivity test. For sorption kinetics, pH 6.0 was used. Why pH 6.0? Why not pH 8.0? considering that this material will be potentially applied in uranium extraction from seawater. Besides, pH-dependence on the sorption should be assessed in helping to understand the sorption mechanism.

3. the author needs to compare and contrast the H2BHT polymeric adsorbent performance with unmodified and H3IDO counterpart to establish that the new sorbent is superior. And also, the current results does not provide enough evidences that could substantiate why the H2BHT polymeric adsorbent is superior to other existing sorbents, only because of the selectivity towards uranium (VI) versus vanadium (V)? "the uranium concentration dropped significantly from 10 ppm to ~0.15 ppm despite the high concentration of competing vanadium (V) species in solution" how about the change of vanadium concentration? this is a key point.

4. In this work, the binding of U(VI) on the surface of the H2BHT polymeric adsorbent were not characterized and analysed. Although the solution chemistry of H2BHT ligand with U(VI) was well studied, upon anchored on the solid surface, the case will be very different.

5. It will be interesting to show that the new material could be regenerated and used to sorb U(VI) with the same efficiency which is needed to evaluate its economic and environmental feasibility.

6. In Figure 3e, the points between 8 h and 25 h seem to be missed. These points are essential for determining the equilibrium time.

Reviewer #2 (Remarks to the Author):

This study entitled by "Siderophore-inspired chelator hijacks uranium from aqueous medium" relates to a selective recovery of uranium from seawater in a presence of vanadium, which is hard to be separated. The authors employed comprehensive chemical techniques, including synthetic, spectroscopic and computational methods, and achieved an effective separation of uranium from vanadium by using H2BHT ligand. It was noteworthy that the efficient selectivity of H2BHT material can be supported by the donation affinity and complexation behavior of molecular H2BHT toward metal ions. This fundamental concern leads to not only understanding a correlation between metal-ligand bonding property and separation behavior but also shedding a light to an exhaustion of natural uranium sources which is a problem in nuclear power generation field. Considering both my opinion mentioned above and several papers relating to uranium coordination science published in Nature Communications, for examples, Ashley et al., 9, 2097 (2018), Qi et al., 9, 1644 (2018), I judge that this manuscript is suitable to be published in Nature Communications. However, there are several unclear points for a choice of model ligand, H2BHT, explanation of origin in uranium/vanadium selectivity, and minor points. I recommend this manuscript to be published after the authors addressed and clarified the following major comments.

General comments:

i) The authors employed H2BHT ligand (Fig. 1c) as siderophore-inspired model molecule. This point was not understandable for me. It was easy to understand that H2BHT ligand is considered as an analogue and improvement of H3IDO ligand (Fig. 1a), possessing the highest adsorption capacity to uranium ion, because a couple of ligands are expected to work as tridentate chelates which consist of two hydroxyl oxygen-donor and one nitrogen-donor atoms. However, a donation moiety of Ferrichrome (Fig. 1b) has a hydroxamic group, consisting of one hydroxyl oxygen donor and one carbonyl oxygen donor. In that case, acetohydroxamic acid, which was employed as an analogue of desferrioxamine B in Guerard et al., ChemComm, 49, 1002 (2013), is more suitable than H2BHT. Ferrichrome is difficult to be considered as one of root compounds of H2BHT molecule. The authors should explain reasonably about the choice of H2BHT as the chemical analogue of Ferrichrome.

ii) This study achieved a selective separation of uranium from vanadium by using polymeric H2BHT material. This might be one of critical points in this study, because H2BHT presents a stronger ability from a viewpoint of selectivity of uranium than H3IDO ligand (although H3IDO shows the highest separation capacity). In order to emphasize the selectivity of H2BHT toward uranium, the authors should mention an origin of the higher selectivity of H2BHT than H3IDO in detail, for example, by the aid of electron density analyses using density functional theory. I strongly recommend that this point is improved to give a clearer description of metal-ligand interaction, because the principal objective of this study is thought to be in coordination chemistry.

Minor comments:

i) In manuscript p.6 1130-134, the difference in the composition of the vanadium complexes between H2BHT and H3IDO systems was mentioned. This difference was indicated to originate the difference in the values of beta. As shown in the values of uranium systems, are these values comparable to the results of DFT calculations?

ii) In supplementary p.11, ΔG_g might include thermal corrections, but it was unclear that definite contributions are contained in the thermal corrections. Please clarify this point.

iii) In supplementary p.11, the relationship between ΔG_{aq} and log beta was not mentioned, although the values were tabulated in manuscript. The authors should describe the procedure in a proper place.

Reviewer #3 (Remarks to the Author):

The manuscript reports on theoretical and experimental studies of 2,6-bis[hydroxy(methyl)amino]-1,3,5-triazine (H2BHT) tridentate ligands for sequestration of UO₂ cations at low ppm concentrations. This process is of substantial interest for extraction of uranyl cations from sea water to provide essentially limitless supply of fuel for nuclear energetics. Authors report a high selectivity of these ligands toward uranyl vs. vanadium(V) cations in solutions in comparison to known hydroxamate and dioxime ligands which is important for potential applications for the extraction. They demonstrated that the H2BHT functionality can be easily used for preparation of polymeric adsorbents possessing high selectivity toward uranyl cations and the ability for easy recycling.

Publication is recommended after addressing the following issues:

Title: The synthesized chelate ligands are quite far away from siderophores so I'd consider removing "siderophore-inspired" from the title

Page 1 Line 13. Siderophores are not necessary hydroxamate based and hydroxamates are not necessary siderophores.

Page 2 Line 59 Siderophores are not necessary hydroxamate based and hydroxamates are not necessary siderophores. The hydroxamate based ones are bis- and tris-hydroxamates small molecules and not polyhydroxamic acids

Page 8 Line 172. Use a systematic name N,N-diisopropylethylamine instead of "Hünig's base"

Page 9 Fig 3d The sorption isotherm does not allow seeing uptake capacities and equilibrium concentrations for the most interesting experiments at the lowest concentrations of uranyl cations. These should be added to the plot or organized in a separate table.

The reported selective adsorption of uranyl cations in the presence of vanadium(V) is very encouraging demonstration of superior selectivity of BHT type ligands in comparison to glutarimide-dioxime ligands. However, interference of iron(III) vs uranyl cations appears to be more serious problem and should be discussed in that context.

Reviewer #1:

General Comments:

The reviewer finds the current manuscript (ms) has many deficiencies that need to be addressed. Although the authors highlight the first preparation of uranyl-selective adsorbent based on H₂BHT artificial siderophore ligand, the characterization and the sorption study are fairly routine and oversimplified.

Response to General Comments: We highly value the reviewer's constructive suggestions and the opportunity to address the raised comments.

Specific Comments:

1. How about the nature of the surface modification? The current characterizations can not support that H₂BHT ligand was modified by chemical binding?

Response:

We appreciate reviewer's comments; however, we would like to emphasize that the structure and functionalization of the fibers is fully supported by comparing $\delta^{13}\text{C}$ CP/MAS solid-state NMR spectrum of the final H₂BHT-functionalized fiber with the spectrum of the H₂BHT small molecule (Figure 3c in the main text). In addition, we obtained IR-spectra of the intermediates and the final H₂BHT-functionalized fiber (Supplementary Figure 8). They were found to be consistent with the expected infra-red absorption pattern of the corresponding functional groups, indicating that the H₂BHT ligand was successfully incorporated into the fiber. To further prove the existence of nitrogen-containing functional groups, we performed an EDX mapping of the H₂BHT-functionalized fiber. This analysis revealed a fairly homogeneous distribution of the nitrogen atoms within the adsorbent material (Supplementary Figure 10), as would be expected from the functionalization of the initial starting material—polyacrylic acid-grafted polyethylene fibers—with H₂BHT moieties.

2. The details on the sorption experiments were not fully provided. The solution pH, for example, was not indicated in sorption isotherm and selectivity test. For sorption kinetics, pH 6.0 was used. Why pH 6.0? Why not pH 8.0 ? considering that this material will be potentially applied in uranium extraction from seawater. Besides, pH-dependence on the sorption should be assessed in helping to understand the sorption mechanism.

Response:

We thank the reviewer for this comment. We have revised the main text and supplementary information to include the pH 6 of all solutions used in the adsorption and selectivity tests. We have added the following sentences to the main text [page 9]:

The adsorption isotherm (pH of the solution was optimized at 6, Supplementary Fig. 12) was obtained to assess the overall capacity of the adsorbent by allowing it to equilibrate with solutions containing varying UO_2^{2+} concentrations ranging from 1 to 250 ppm and a phase ratio of 0.25 mg mL^{-1} .

[page 9 (bottom)]:

The adsorbent sample (6.5 mg) was stirred overnight in 200 mL solution (pH 6) with equal concentrations (10 ppm) of uranium ($\text{UO}_2(\text{NO}_3)_2 \cdot 6\text{H}_2\text{O}$) and vanadium (Na_3VO_4).

and the supplementary information [page 23; Supplementary Methods section]:

The pH levels of all solutions used in sorption isotherm, sorption kinetics, and selectivity studies were adjusted to 6.0 by HNO_3 or NaOH aqueous solution.

The rationale for using pH 6.0 of the solution was to maintain the consistency across the field. Based on our experience, the hydrolysis of uranium(VI) and vanadium(V) ions (using ppm concentrations) becomes severe in basic solution at high pH value, which may result in the precipitation of the ions. Consequently, this creates difficulties for measuring the uranyl/vanadium concentrations precisely at $\text{pH} > 6$. Furthermore, we studied the pH dependence of H_2BHT adsorbent uptake capacities as requested by the reviewer. Figure 12 of the Supplementary Information shows that the highest U adsorption is achieved at pH 6-10, with the adsorption capacity at pH 6 being very close to the adsorption maximum.

3. the author needs to compare and contrast the H_2BHT polymeric adsorbent performance with unmodified and H_3IDO counterpart to establish that the new sorbent is superior. And also, the current results does not provide enough evidences that could substantiate why the H_2BHT polymeric adsorbent is superior to other existing sorbents, only because of the selectivity towards uranium (VI) versus vanadium (V)? “the uranium concentration dropped significantly from 10 ppm to ~0.15 ppm despite the high concentration of competing vanadium (V) species in solution” how about the change of vanadium concentration? this is a key point.

Response:

We have added a table to the Supplementary Information (Supplementary Table 9) with the data on the published performance of the conventional amidoxime (H_3IDO) sorbent under similar conditions at $\text{pH} = 6$. As can be seen, the H_3IDO adsorbent is consistently more selective for vanadium (V) ions over uranyl, despite almost equal starting concentrations of vanadium ($[\text{V}] = 5.8 \times 10^{-5} \text{ M}$) and uranyl ($[\text{U}] = 3.2 \times 10^{-5} \text{ M}$) used in the sorption experiment. This behavior can be explained by the ability of the H_3IDO ligand to

form very stable non-oxido V complex with a high stability constant value (please see the discussion in the main text and ref. 22 in the main text; Ivanov, et al. *Nat. Commun.* **8**, 1560 (2017)). Therefore, we believe that the small molecule binding constants reported in our study are particularly revealing, since they can explain the superior selectivity of the developed H₂BHT adsorbent for uranium over vanadium. In fact, we provide the direct comparison of the thermodynamic parameters addressing the binding between the novel H₂BHT and conventional H₃IDO moieties. We further provide evidence of the superior selectivity of the H₂BHT towards uranium over vanadium, as opposed to H₃IDO, which clearly has a preference toward vanadium ions.

To address the second part of the reviewer's comment and to confirm the observed superior uranium vs. vanadium selectivity of H₂BHT polymer, we have traced the vanadium concentration in solution by performing additional competition experiments using the mixed solution of vanadium ([V]=1.18×10⁻⁴ M) and uranyl ([U]=0.42×10⁻⁵ M) at [V]/[U] ~3/1 concentration ratio (excess of vanadium ions). The results in Supplementary Table 9 show that the starting vanadium concentration didn't change after adding the H₂BHT adsorbent. Based on our data, we can conclude that our novel material exhibits superior selectivity towards uranium over vanadium. We have added the following sentences to the Supporting Information [Supplementary Methods section]:

Additional competitive adsorption experiments were performed to confirm the observed selectivity in the presence of high concentration of vanadium (V) species in solution. A solution was made by mixing UO₂(NO₃)₂·6H₂O and Na₃VO₄ at [V]/[U] ~ 3/1 concentration ratio ([V] = 1.18×10⁻⁴ M; [U] = 0.42×10⁻⁴ M). 50 mg of adsorbent was added to 400 mL of the mixed metal solution, this was then stirred overnight. The solutions were filtered through a 0.45 μm membrane filter, and the filtrate was analyzed by ICP for the remaining vanadium concentrations (reported in Supplementary Table 8).

And [Supplementary Note 3]:

The vanadium (V) concentration remained unchanged (Supplementary Table 8), indicating no adsorption of vanadium ions by the H₂BHT polymer. This behavior can likely be explained by the difference in the formation constant values for uranium vs. vanadium complexes with the H₂BHT functional group. According to our theoretical and experimental studies on small molecule analogs, vanadium binding constant for H₂BHT (log β = 17.9)¹³ is much lower than the uranyl binding strength with two H₂BHT ligands (log β = 41.9, Table 1 in the main text). Therefore, vanadium would not likely compete with uranium for H₂BHT active sites, explaining impressive selectivity of the developed H₂BHT-functionalized polymer toward uranyl over vanadium species in aqueous medium.

4. In this work, the binding of U(VI) on the surface of the H₂BHT polymeric adsorbent were not characterized and analysed. Although the solution chemistry of H₂BHT ligand with U(VI) was well studied, upon anchored on the solid surface, the case will be very different.

Response:

We highly value the reviewer's comment. We have performed additional experiments to address this point. Specifically, solid state NMR and IR data strongly support the presence of H₂BHT functionality on the fibers. XPS data of the uranium-contacted fibers reveal the expected peaks corresponding to uranium 4f transitions (Supplementary Figure 9). In addition, the IR spectrum of the uranium contacted fiber exhibits the expected O=U=O bond stretching in 900cm⁻¹ region (Supplementary Figure 11). Finally, EDX mapping of the uranium-contacted fiber shows the presence of all the elements expected on the surface of the fiber, i.e. C, N, O, U (Supplementary Figure 10). This is a very strong indication of the existence of nitrogen-containing groups, i.e. H₂BHT, and that indeed they are responsible for the binding of uranyl cation. We have added 3 new figures (Supplementary Figures 9-11) to the Supplementary Information and the following sentences to the main text [page 9 (top)]:

With the H₂BHT polymer adsorbent in hand, we investigated the uranium (VI) binding by this material using a variety of spectroscopic techniques, including X-ray photoelectron spectroscopy (XPS) and elemental distribution mapping via energy-dispersive X-ray (EDX) spectroscopy analysis. Uranium inclusion within the adsorbent is evident from the characteristic U 4f signals in the XPS spectra of the uranyl reacted polymer sample (Supplementary Fig. 9). The EDX elemental mapping results (Supplementary Fig. 10) confirm the existence of uranium species on the surface of the H₂BHT functionalized polymer with a homogeneous distribution of the extracted uranium throughout the material. Furthermore, FT-IR spectrum of the uranyl contacted sample (Supplementary Fig. 11) exhibits a significant red-shifted peak that corresponds to the antisymmetric stretch vibrational mode of [O=U=O]²⁺, indicating strong interactions between uranyl and the H₂BHT functional groups in the developed polymeric adsorbent.

5. It will be interesting to show that the new material could be regenerated and used to sorb U(VI) with the same efficiency which is needed to evaluate it's economic and environmental feasibility.

Response:

We thank the reviewer for this comment. We performed our regeneration tests 3 times. Indeed, the H₂BHT-functionalized adsorbent can be fully regenerated using sodium carbonate solution as a strip solution and shows similar performance even after 3 cycles (Supplementary Table 7 and Supplementary Note 3). However, we would like to point out that the focus of the present communication is not on the economic feasibility of this material with respect to its performance (this would have to be analyzed and further developed by process engineers), but on the proof-of-principle study and fundamental aspects of binding affinity and selectivity of the materials as it pertains to small molecule ligand. We would like to

specifically emphasize the novelty of our strategy towards the synthesis of this H₂BHT-based material and the implications that this approach will have on the development of other selective polymeric systems with the tailor-made functional groups.

6. In Figure 3e, the points between 8 h and 25 h seem to be missed. These points are essential for determining the equilibrium time.

Response:

We appreciate the reviewer's comment. We have performed additional tests to address this issue. The modified Figure 3e in the main text now contains additional data in the interval between 8h and 25h.

We would like to thank the reviewer for valuable and constructive suggestions, which significantly helped improve the quality of our manuscript.

Reviewer #2:

General Comments:

This study entitled by "Siderophore-inspired chelator hijacks uranium from aqueous medium" relates to a selective recovery of uranium from seawater in a presence of vanadium, which is hard to be separated. The authors employed comprehensive chemical techniques, including synthetic, spectroscopic and computational methods, and achieved an effective separation of uranium from vanadium by using H₂BHT ligand. It was noteworthy that the efficient selectivity of H₂BHT material can be supported by the donation affinity and complexation behavior of molecular H₂BHT toward metal ions. This fundamental concern leads to not only understanding a correlation between metal-ligand bonding property and separation behavior but also shedding a light to an exhaustion of natural uranium sources which is a problem in nuclear power generation field.

Considering both my opinion mentioned above and several papers relating to uranium coordination science published in Nature Communications, for examples, Ashley et al., 9, 2097 (2018), Qi et al., 9, 1644 (2018), I judge that this manuscript is suitable to be published in Nature Communications. However, there are several unclear points for a choice of model ligand, H₂BHT, explanation of origin in uranium/vanadium selectivity, and minor points. I recommend this manuscript to be published after the authors addressed and clarified the following major comments.

Response to General Comments: We greatly appreciate the very positive evaluation of our work by the reviewer.

Specific Comments:

i) The authors employed H₂BHT ligand (Fig. 1c) as siderophore-inspired model molecule. This point was not understandable for me. It was easy to understand that H₂BHT ligand is considered as an analogue and improvement of H₃IDO ligand (Fig. 1a), possessing the highest adsorption capacity to uranium ion, because a couple of ligands are expected to work as tridentate chelates which consist of two hydroxyl oxygen-donor and one nitrogen-donor atoms. However, a donation moiety of Ferrichrome (Fig. 1b) has a hydroxamic group, consisting of one hydroxyl oxygen donor and one carbonyl oxygen donor. In that case, acetohydroxamic acid, which was employed as an analogue of desferrioxamine B in Guerard et al., ChemComm, 49, 1002 (2013), is more suitable than H₂BHT. Ferrichrome is difficult to be considered as one of root compounds of H₂BHT molecule. The authors should explain reasonably about the choice of H₂BHT as the chemical analogue of Ferrichrome.

Response:

We appreciate the reviewer's comments. The reason why we consider H₂BHT ligand to be a siderophore-mimetic, is primarily due to its function and functionality rather than a direct structural similarity, i.e. H₂BHT—an aromatic hydroxylamine chemically behaves similarly to polyphenol- and polyhydroxamate-based siderophores in its propensity towards binding iron ions. One key feature of siderophores-naturally occurring iron transporters- is their very high affinity towards this metal. Melman et al. has demonstrated that H₂BHT is an efficient Iron chelator (ref. 29,30 in the main text; Melman, et al. *Chem. Commun.* 5319–5321 (2005); Melman, et al. *Dalton Trans.* 1285–1293 (2006)). Indeed, H₃IDO could be considered siderophore-mimetic in its function as well, since it too exhibits a very high binding affinity towards iron in aqueous solution (Rao, et al. *Dalton Trans.* 42, 14621–14627 (2013)).

ii) This study achieved a selective separation of uranium from vanadium by using polymeric H₂BHT material. This might be one of critical points in this study, because H₂BHT presents a stronger ability from a viewpoint of selectivity of uranium than H₃IDO ligand (although H₃IDO shows the highest separation capacity). In order to emphasize the selectivity of H₂BHT toward uranium, the authors should mention an origin of the higher selectivity of H₂BHT than H₃IDO in detail, for example, by the aid of electron density analyses using density functional theory. I strongly recommend that this point is improved to give a clearer description of metal-ligand interaction, because the principal objective of this study is thought to be in coordination chemistry.

Response:

We thank the reviewer for this valuable suggestion. We agree that in order to elucidate the origin of the strong uranyl binding by the H₂BHT and H₃IDO functionalities it is important to investigate ligand-uranyl orbital interactions. Therefore, we performed chemical bonding analysis of the respective neutral 1:1

UO₂BHT and UO₂HIDO complexes using the natural bond orbital (NBO) method. NBO analysis provides a good quantitative description of interatomic and intermolecular interactions in accordance with the basic Pauling–Slater–Coulson representations of bond polarization and hybridization. The results of our chemical bonding analysis are summarized in Supplementary Table 3. As expected, NBO confirmed very similar uranyl binding strengths of H₂BHT and H₃IDO. We have added the following sentences to the main text [page 4 (bottom)]:

Natural bond orbital (NBO) analysis further confirms strong uranyl binding by both functionalities, revealing effective metal-ligand interactions via dative σ -bonds for the neutral 1:1 H₂BHT and H₃IDO uranyl complexes (Supplementary Table 3).

and to the Supplementary Information [Supplementary Methods section]:

Chemical bonding analysis was performed with the natural bond orbital (NBO) method^{11,12} at the M06/SSC/6-311++G(d,p) level. NBO analysis provides a good quantitative description of interatomic and intermolecular interactions in accordance with the basic Pauling-Slater-Coulson representations of bond polarization and hybridization.^{11,12} The donor-acceptor interaction energy in the NBOs was estimated via second-order perturbation theory analysis of the Fock matrix.¹¹ For each donor orbital (i) and acceptor orbital (j), the stabilization energy $E^{(2)}$ associated with $i \rightarrow j$ delocalization is given by:

$$E_{i,j}^{(2)} = -o_i \frac{\langle i | \hat{F}_{(i,j)} | j \rangle^2}{\epsilon_j - \epsilon_i} \quad (3)$$

where o_i is the donor orbital occupancy, $\hat{F}_{(i,j)}$ is the Fock operator, and ϵ_i and ϵ_j are the orbital energies.

We would like to point out that the greater selectivity of H₂BHT for uranium over vanadium species originates from the high stability constant values of the H₂BHT uranyl complexes compared to that of H₂BHT vanadium complexes. In contrast, H₃IDO ligand is known to form very stable and rare 1:2 non-oxido vanadium complex with the highest stability constant value ever observed for the V⁵⁺ species (ref. 22 in the main text; Ivanov, et al. *Nat. Commun.* **8**, 1560 (2017)). The inability of H₂BHT to form analogous 1:2 non-oxido vanadium complex was also confirmed by additional DFT calculations (please see our response below for more details).

Minor Comments:

i) In manuscript p.6 1130-1134, the difference in the composition of the vanadium complexes between H₂BHT and H₃IDO systems was mentioned. This difference was indicated to originate the difference in the values of beta. As shown in the values of uranium systems, are these values comparable to the results of DFT calculations?

Response:

Although experimental $\log \beta$ values of the H₂BHT vanadium complexes have already been reported by Nikolakis et al. (ref. 33 in the main text), we have applied our DFT-based approach to predict the stability constants for the vanadium (V) complexes with H₂BHT as requested by the reviewer. The computational results were found to be in good agreement with experimental data. Moreover, we computationally predicted $\log \beta$ for a hypothetical 1:2 V(BHT)₂⁺ complex. The simulated speciation diagram based on the theoretically predicted $\log \beta$ values of the vanadium complexes indeed doesn't show 1:2 V(BHT)₂⁺ complex formation, which is suppressed by the more stable 1:1 VO₂(BHT)⁻ species (Supplementary Figure 2). We have added Supplementary Figure 2, Supplementary Table 5, and the following sentences to the Supplementary Information [Supplementary Note 1]:

Supplementary Note 1. Although there is no experimental evidence for the displacement of the V=O oxido bonds by H₂BHT, we considered the possible formation of a hypothetical 1:2 non-oxido V(BHT)₂⁺ complex to provide a complete picture of the V(V) complexation by the H₂BHT ligand. First, we applied our computational protocol¹⁰ to predict $\log \beta^{theor}$ values of the conventional 1:1 dioxovanadium complexes. The obtained results ($\log \beta^{theor}$ [VO₂(BHT)⁻] = 17.3; $\log \beta^{theor}$ [VOOH(BHT)] = 19.1 with respect to H₂VO₄⁻ species) are in good agreement with the experimental data reported relative to H₂VO₄⁻ species by Nikolakis et al.¹³ ($\log \beta^{expt}$ [VO₂(BHT)⁻] = 17.87 (1); $\log \beta^{expt}$ [VOOH(BHT)] = 19.39 (6). The corresponding $\log \beta^{theor}$ for the non-oxido V(BHT)₂⁺ complex can be obtained by a combination of the reactions for which ΔG_{aq} is experimentally known or was assessed using DFT calculations. The complexation free energy, ΔG_{aq8} , and subsequently $\log \beta^{theor}$ for the formation of the V(BHT)₂⁺ complex was found by combination of the following reactions ($\Delta G_{aq9} = 2\Delta G_{aq6} + 2G_{aq7} + \Delta G_{aq8}$):

where ΔG_{aq6} is experimentally known value from Supplementary Table 5 (at 25 °C and $I = 0$ M), ΔG_{aq7} was determined using our methodology from supplementary reference 10, and the free energy, ΔG_{aq8} , was calculated at the M06/SSC/6-311++G(d,p) level of theory. The obtained ΔG_{aq8} value of +13.33 kcal mol⁻¹ indicate a thermodynamically unfavorable process. This is in direct contrast to the H₃IDO ligand, for which ΔG_{aq} of a similar reaction (VOOH(HIDO)⁻ + VOOH(HIDO)⁻ \rightleftharpoons V(IDO)₂⁻ + H₂VO₄⁻) was calculated to be -0.11 kcal mol⁻¹ at the same level of theory. In addition, the generated species distribution diagrams (Supplementary Fig.2), constructed by incorporating $\log \beta^{theor}$ of the V(V)/H₂BHT complexes (Supplementary Table 5) along with the experimental hydrolysis constants for mononuclear vanadium(V) species, show that only 1:1 vanadium complexes are the dominant species at both [H₂BHT]/[V] = 1/1 and [H₂BHT]/[V] = 100/1 concentration ratios over the pH range of ~3–10. It is also worth noting that the speciation data suggest the feasibility of removing all vanadium from the H₂BHT fiber by increasing pH

of the solution, which contrasts with the H₃IDO-functionalized adsorbent materials (see reference 22 in the main text).

and to the main text [page 6 (top)]:

The inability of H₂BHT to form a hypothetical 1:2 non-oxido vanadium complex V(BHT)₂⁺ was further supported by DFT calculations (Supplementary Note 1 and Supplementary Table 5). The simulated speciation diagrams in Supplementary Fig. 2 indicate that there is no V(BHT)₂⁺ complex formation over the entire pH range even in the presence of a large excess of the ligand.

ii) In supplementary p.11, ΔG_g might include thermal corrections, but it was unclear that definite contributions are contained in the thermal corrections. Please clarify this point.

Response:

We have clarified our computational procedures by providing more information in the Supplementary Methods section:

Frequency calculations were performed at the B3LYP/SSC/6-31+G(d) level to ensure that geometries (optimized at the same level) were minima and to compute zero-point energies and thermal corrections to Gibbs free energy, G_{corr} , within the harmonic oscillator approximation using standard formulae based on statistical thermodynamics⁵ ($G_{\text{corr}} = H_{\text{corr}} - TS_{\text{tot}}$, where enthalpy ($H_{\text{corr}} = E_{\text{tot}} + k_{\text{B}}T$) and entropy (S_{tot}) include translational, rotational, vibrational, and electronic contributions to the internal thermal energy ($E_{\text{tot}} = E_{\text{t}} + E_{\text{r}} + E_{\text{v}} + E_{\text{e}}$) and entropy ($S_{\text{tot}} = S_{\text{t}} + S_{\text{r}} + S_{\text{v}} + S_{\text{e}}$), respectively). The obtained values of zero-point energy and thermal corrections to the total energy of species were used to calculate gas-phase free energy change, ΔG_g^o , in accord with equation 1.

iii) In supplementary p.11, the relationship between ΔG_{aq} and $\log \beta$ was not mentioned, although the values were tabulated in manuscript. The authors should describe the procedure in a proper place.

Response:

We have provided more information regarding our computational approach for predicting $\log \beta$:

More specifically, this approach utilizes a thermodynamic cycle scheme involving the calculation of the gas-phase free energies and the change in free energy upon transfer of 1 mole of a species from the gas to the aqueous phase under standardized conditions. Once the various free energy terms of the cycle are calculated using quantum chemical methods, then the change in free energy for the aqueous reaction (ΔG_{aq}) can be determined. The $\log \beta$ values are obtained from the relation of $\log \beta$ to ΔG_{aq} by the following equation:

$$\log \beta = -\frac{\Delta G_{aq}}{2.303 RT} \quad (2)$$

Finally, $\log \beta^{theor}$ values in the main text are reported after applying the corresponding regression equations⁹ to the calculated $\log \beta$. These regression equations (1:1 and 1:2 uranyl complexes: [$\log \beta^{expt} = 0.5693 \times \log \beta^{calc}$]⁹; 1:2 uranyl complexes carrying an excess negative charge: [$\log \beta^{expt} = 0.6498 \times \log \beta^{calc} - 7.7565$]⁹; 1:1 vanadium (V) complexes: [$\log \beta^{expt} = 0.390 \times \log \beta^{calc} - 1.583$]¹⁰) were in turn obtained from the extensive correlations between quantum mechanical calculations of $\log \beta$ and available experimental data.^{7,8,10}

Reviewer #3:

General Comments:

The manuscript reports on theoretical and experimental studies of 2,6-bis[hydroxy(methyl)amino]-1,3,5-triazine (H2BHT) tridentate ligands for sequestration of UO₂ cations at low ppm concentrations. This process is of substantial interest for extraction of uranyl cations from sea water to provide essentially limitless supply of fuel for nuclear energetics. Authors report a high selectivity of these ligands toward uranyl vs. vanadium(V) cations in solutions in comparison to known hydroxamate and dioxime ligands which is important for potential applications for the extraction. They demonstrated that the H2BHT functionality can be easily used for preparation of polymeric adsorbents possessing high selectivity toward uranyl cations and the ability for easy recycling. Publication is recommended after addressing the following issues.

Response to General Comments: We highly appreciate the reviewer's positive evaluation of our work.

Specific Comments:

Title: The synthesized chelate ligands are quite far away from siderophores so I'd consider removing "siderophore-inspired" from the title

Response:

We appreciate the reviewer's comment. However, we would like to point out to the functional similarities between the classes of siderophores and H₂BHT ligand. We consider H₂BHT ligand a siderophore-mimetic due to its function and functionality rather than a direct structural similarity, i.e. H₂BHT—an aromatic hydroxylamine chemically behaves similarly to polyphenol- and polyhydroxamate-based siderophores in its propensity towards binding iron ions. One key feature of siderophores-naturally occurring iron transporters-is their very high affinity towards this metal. Melman et al. has demonstrated

that H₂BHT is an efficient Iron chelator (ref. 29,30 in the main text; Melman, et al. *Chem. Commun.* 5319–5321 (2005); Melman, et al. *Dalton Trans.* 1285–1293 (2006)).

Page 1 Line 13. Siderophores are not necessary hydroxamate based and hydroxamates are not necessary siderophores.

Response:

We absolutely agree with the reviewer. We have changed the language used in the manuscript to remove any ambiguity.

Page 2 Line 59 Siderophores are not necessary hydroxamate based and hydroxamates are not necessary siderophores. The hydroxamate based ones are bis- and tris-hydroxamates small molecules and not polyhydroxamic acids

Response:

We thank the reviewer for this comment. We have modified the text in the manuscript to reflect this [page 3 (top)]:

Siderophores, a class of naturally occurring (Fig 1b) nitrogen- and oxygen-based donor group containing chelating compounds for iron sequestration present in bacteria and fungi that exhibit some of the highest binding affinities towards the metal, have long been recognized as potential ligands for f-block elements.

Page 8 Line 172. Use a systematic name N,N-diisopropylethylamine instead of "Hünig's base"

Response:

We have changed the term "Hünig's base" to N,N-diisopropylethylamine, per reviewer's suggestion.

Page 9 Fig 3d The sorption isotherm does now allow seeing uptake capacities and equilibrium concentrations for the most interesting experiments at the lowest concentrations of uranyl cations. These should be added to the plot or organized in a separate table.

Response:

Following the reviewer's suggestion, we have added Supplementary Table 6:

As was envisioned from small molecule studies, the H₂BHT adsorbent material has high affinity towards uranyl ion in aqueous solution with the maximum uptake capacity of 105 mg g⁻¹ with an equilibrium concentration of 205 ppm (Supplementary Table 6).

The reported selective adsorption of uranyl cations in the presence of vanadium(V) is very encouraging demonstration of superior selectivity of BHT type ligands in comparison to glutaroimide-dioxime ligands. However, interference of iron(III) vs uranyl cations appears to be more serious problem and should be discussed in that context.

Response:

We appreciate the comments provided by the reviewer. Indeed, under the circumstances where iron ion concentration is comparable to that of uranium, based on the binding constants, the H₂BHT ligand would preferentially bind to iron. However, one important distinction that we would like to emphasize is that, unlike vanadium, iron concentration in the aqueous environment is low due to a number of reasons. First, solubility of iron oxide/hydroxide/carbonate/silicate salts that could be potentially present on aqueous environment is very low, in addition, living organisms constantly compete to sequester any remaining soluble iron. For instance, polyamidoxime adsorbents containing mostly glutaroimide-dioxime (H₃IDO) functional groups adsorb only ~1 g of Iron after 21 days of contact with seawater, while the uptake of Uranium (~3 g) and Vanadium (~7.5 g) is much higher (ref. 22 in the main text; Ivanov, et al. *Nat. Commun.* **8**, 1560 (2017)). However, the stability constants for Iron complexes with H₃IDO (e.g. $\log \beta$ [Fe(HIDO)₂]⁻ = 36.02) are generally higher than those for Uranyl-H₃IDO (e.g. $\log \beta$ [UO₂(HIDO)₂]²⁻ = 27.5) (Rao, et al. *Dalton Trans.* 42, 14621–14627 (2013)).

Additionally, in case of polyamidoxime (H₃IDO) adsorbent, iron complexes could be efficiently hydrolyzed (stripped from the fiber) under acidic pH conditions, while vanadium complexes are so stable that the decomposition and chemical destruction of the glutaroimide-dioxime ligand itself is taking place under the conditions that lead to the hydrolysis of the complex. As a consequence, removal of the adsorbed vanadium presents a much bigger problem towards recyclability of the adsorbent than any other metal ions.

Reviewers' comments:

Reviewer #1 (Remarks to the Author):

I would like to thank the authors for addressing my original concerns. I am mostly satisfied with the changes, but I still have a few questions/comments.

1. The H2BHT-functionalized polymeric adsorbent was prepared via five linear steps as indicated in Supplementary Figure 7, constituting the first proof-of-principle synthesis of an artificial siderophore-embedded uranium adsorbent. To substantiate such a synthesis strategy, all the intermediates should be fully characterized. Besides FTIR, for example, $\delta^{13}\text{C}$ CP/MAS solid-state NMR spectra should be provided.

2. All the current characterizations reveal the presence of H2BHT functionality on the polymeric adsorbent, and the presence of uranyl cations following the adsorption. However, the interaction between the adsorbent and uranyl cations are poorly analysed. EXAFS may be an effective tool, and the results can be compared with the solution chemistry of H2BHT ligand with U(VI).

3. According to the thermodynamic data, U(VI) precipitation occurs mainly at pH 6-7. Beyond this range, e.g. pH 8.3, as is generally the case for seawater, uranyl carbonate complexes with negative charge become dominant species. Given that the polymeric adsorbent are potentially used to extract U(VI) from seawater, it is strongly suggested that the selective recognition of uranium (VI) over vanadium (V) by this adsorbent at pH 9-10 should be tested. The case may be very different.

Reviewer #2 (Remarks to the Author):

I appreciate authors' revisions and point-by-point response of a manuscript entitled "Siderophore-inspired chelator hijacks uranium from aqueous medium" and believe that the revised manuscript is convincing to be published in Nature Communications after addressing previous comments. This revision gives clearer explanation into complexation reactions between vanadium(V)/uranium(VI) ions and H2BHT/H3IDO ligands using DFT calculations and the electron population analysis. The authors performed the Gibbs energy calculations in the complexation reactions of $[\text{V}(\text{BHT})_2]^+$, which is experimentally unfavorable, and $[\text{V}(\text{IDO})_2]^-$ complex, which is experimentally favorable. The DFT calculation of the standard Gibbs energy differences, ΔG , of the complexes, $[\text{V}(\text{BHT})_2]^+$ by $2[\text{VOOH}(\text{BHT})] \rightleftharpoons [\text{V}(\text{BHT})_2]^+ + \text{H}_2\text{VO}_4^-$ ($\Delta G = +13.33 \text{ kcal mol}^{-1}$), and $[\text{V}(\text{IDO})_2]^-$ by $2[\text{VOOH}(\text{HIDO})]^- \rightleftharpoons [\text{V}(\text{IDO})_2]^- + \text{H}_2\text{VO}_4^-$ ($\Delta G = -0.11 \text{ kcal mol}^{-1}$), reproduced the unfavorable nature of $[\text{V}(\text{BHT})_2]^+$ and favorable nature of $[\text{V}(\text{IDO})_2]^-$. This result indicated to be one evidence of the higher U(VI) selectivity over V(V) with H2BHT ligand compared to that with H3IDO ligand, because a stabilization of $[\text{VOOH}(\text{BHT})]$ by one more complexation of BHT²⁻, which is gained in the case of $[\text{VOOH}(\text{IDO})]^-$, is not expected. It leads to the relatively high stability of $[\text{UO}_2(\text{BHT})]$ complex compared to the vanadium complex. Additional result of natural bond orbital (NBO) analysis for uranyl species, $[\text{UO}_2(\text{L})(\text{H}_2\text{O})_2]$ (L = BHT²⁻ and HIDO²⁻), indicated that the coordination bonding properties for both complexes are quite similar each other. A couple of computational results succeeded in explaining the reason why H2BHT ligand has higher uranium(VI)/vanadium(V) selectivity than H3IDO ligand.

In this revised version of your manuscript, the high selectivity of H2BHT as one of the main claim of this manuscript could be strengthened by combining experimental and computational techniques. From the above reasons, I recommend the manuscript to be published in Nature Communications without change.

Reviewer #3 (Remarks to the Author):

Page 9 Fig 3d The sorption isotherm does not allow seeing uptake capacities and equilibrium concentrations for the most interesting experiments at the lowest concentrations of uranyl cations. This figure can be either added to the plot or organized in a separate table.

The reported selective adsorption of uranyl cations in the presence of vanadium(V) is a very encouraging demonstration of superior selectivity of BHT type ligands in comparison to glutarimide-dioxime ligands. However, interference of iron(III) vs uranyl cations appears to be a more serious problem, and should be discussed in that context. Data on concentrations of uranium and iron cations in seawater along with dissociation constants of their H₂BHT ligands should be also discussed.

Reviewer #1:

General Comments:

I would like to thank the authors for addressing my original concerns. I am mostly satisfied with the changes, but I still have a few questions/comments.

Response to General Comments: We are happy to know that our revised manuscript has fully addressed the original comments. We also would be happy to answer additional questions/comments raised by the reviewer.

Additional Specific Comments:

1. The H₂BHT-functionalized polymeric adsorbent was prepared via five linear steps as indicated in Supplementary Figure 7, constituting the first proof-of principle synthesis of an artificial siderophore-embedded uranium adsorbent. To substantiate such a synthesis strategy, all the intermediates should be fully characterized. Besides FTIR, for example, δ ¹³C CP/MAS solid-state NMR spectra should be provided.

Response: We have added δ ¹³C CP/MAS solid-state NMR spectra of the key intermediates to the Supplementary Information (please see Supplementary Figure 9).

2. All the current characterizations reveal the the presence of H₂BHT functionality on the polymeric adsorbent, and the presence of uranyl cations following the adsorption. However, the interaction between the adsorbent and uranyl cations are poorly analysed. EXAFS may be an effective tool, and the results can be compared with the solution chemistry of H₂BHT ligand with U(VI).

Response: We have characterized the uranyl binding by H₂BHT-functionalized polymer using state-of-the-art approaches available to us. X-ray photoelectron spectroscopy (XPS) investigations indeed show adsorption of uranium as evidenced by the appearance of the characteristic U 4f signals in the XPS spectra. In addition, FT-IR studies indicate strong interactions between uranyl and the H₂BHT functional groups in the developed polymeric adsorbent. We would like to point out that EXAFS technique is not a unique method to analyze the interactions between the adsorbent and uranyl cations, and as with many other experimental techniques used to probe solution-phase structure, EXAFS data interpretation is only as good as the model used. It is possible that many other plausible structural models could be used to give equally good fits to the EXAFS spectra, leaving sometimes more questions than answers regarding the true nature of interactions between the adsorbent and uranyl species. While we partially agree with the reviewer, we cannot perform EXAFS measurements now, because of the lack of the beam time at the

synchrotron facility and the absence of experts on the EXAFS technique among our coauthors and collaborators. We note, however, that EXAFS measurements will be the focus of our future detailed work on optimizing the adsorption characteristics of the developed H₂BHT-based sorbent material.

3. According to the thermodynamic data, U(VI) precipitation occurs mainly at pH 6-7. Beyond this range, e.g. pH 8.3, as is generally the case for seawater, uranyl carbonate complexes with negative charge become dominant species. Given that the polymeric adsorbent are potentially used to extract U(VI) from seawater, it is strongly suggested that the selective recognition of uranium (VI) over vanadium (V) by this adsorbent at pH 9-10 should be tested. The case may be very different.

Response: We agree with the reviewer that at basic pH (9-10) the adsorption of uranium (VI) vs. vanadium (V) can be different. However, as we pointed out in our original response, uranium species at ppm concentrations are prone to hydrolysis and precipitation from solutions at pH values above 9 (for more details please see the representative paper (Tomažić, B. et al. Precipitation and Hydrolysis of Uranium (VI) in Aqueous Solutions: Uranyl Nitrate-Potassium Hydroxide-Neutral Electrolyte. Croatica Chemica Acta 34, (1962) 41-50)). Therefore, the selectivity tests at pH 9-10 may not be precise. In addition, considering that current industrial emissions of CO₂ are making the oceans more acidic, performing experiments at basic pH has little relevance with real seawater.

We would like to thank the reviewer once again and hope that our manuscript is now suitable for publication.

Reviewer #2:

General Comments:

I appreciate authors' revisions and point-by-point response of a manuscript entitled "Siderophore-inspired chelator hijacks uranium from aqueous medium" and believe that the revised manuscript is convincing to be published in Nature Communications after addressing previous comments. This revision gives clearer explanation into complexation reactions between vanadium(V)/uranium(VI) ions and H₂BHT/H₃IDO ligands using DFT calculations and the electron population analysis. The authors performed the Gibbs energy calculations in the complexation reactions of [V(BHT)₂]⁺, which is experimentally unfavorable, and [V(IDO)₂]⁻ complex, which is experimentally favorable. The DFT calculation of the standard Gibbs energy differences, ΔG, of the complexes, [V(BHT)₂]⁺ by 2[VOOH(BHT)] ⇌ [V(BHT)₂]⁺ + H₂VO₄⁻ (ΔG = +13.33 kcal mol⁻¹), and [V(IDO)₂]⁻ by 2[VOOH(HIDO)]⁻ ⇌ [V(IDO)₂]⁻ + H₂VO₄⁻ (ΔG = -0.11 kcal mol⁻¹), reproduced the unfavorable nature of [V(BHT)₂]⁺ and favorable nature of [V(IDO)₂]⁻. This result indicated to be one evidence of the higher U(VI) selectivity over V(V) with H₂BHT ligand compared to that with H₃IDO ligand, because a

stabilization of [VOOH(BHT)] by one more complexation of BHT²⁻, which is gained in the case of [VOOH(IDO)]⁻, is not expected. It leads to the relatively high stability of [UO₂(BHT)] complex compared to the vanadium complex. Additional result of natural bond orbital (NBO) analysis for uranyl species, [UO₂(L)(H₂O)₂] (L = BHT²⁻ and HIDO²⁻), indicated that the coordination bonding properties for both complexes are quite similar each other. A couple of computational results succeeded in explaining the reason why H₂BHT ligand has higher uranium(VI)/vanadium(V) selectivity than H₃IDO ligand.

In this revised version of your manuscript, the high selectivity of H₂BHT as one of the main claim of this manuscript could be strengthened by combining experimental and computational techniques. From the above reasons, I recommend the manuscript to be published in Nature Communications without change.

Response to General Comments: We are thankful to the reviewer for her/his favorable comments. The reviewer recommended to publish the paper as is.

Reviewer #3:

Additional Specific Comments:

1. Page 9 Fig 3d The sorption isotherm does now allow seeing uptake capacities and equilibrium concentrations for the most interesting experiments at the lowest concentrations of uranyl cations. This figure can be either added to the plot or organized in a separate table.

Response: We have modified Fig 3d (the equilibrium concentration (ppm) of uranyl ions was added for each data point), so that readers can now clearly see the uptake capacities and equilibrium concentrations for the most interesting experiments at the lowest concentrations of uranyl cations. In addition, we have organized Supplementary Table 6 with tabulated experimental results used to plot the sorption isotherm [page 10 (bottom)].

As was envisioned from small molecule studies, the H₂BHT adsorbent material has high affinity towards uranyl ion in aqueous solution with the maximum uptake capacity of 105 mg g⁻¹ with an equilibrium concentration of 205 ppm (experimental data used to plot the uranium sorption isotherm are given in Supplementary Table 6).

2. The reported selective adsorption of uranyl cations in the presence of vanadium(V) is a very encouraging demonstration of superior selectivity of BHT type ligands in comparison to glutarimide-dioxime ligands. However, interference of iron(III) vs uranyl cations appears to be more serious problem, and should be discussed in that context. Data on concentrations of uranium and iron cations in seawater along with dissociation constants of their H₂BHT ligands should be also discussed.

Response: We agree with the reviewer that H₂BHT ligand should possess high binding affinity toward iron(III), since it was first reported by Melman et al. (Chem Commun 5319–5321 (2005)) as a strong

iron(III) chelating agent. Melman reported the formation constant of 1:2 $\text{Fe}^{3+}:\text{H}_2\text{BHT}$ complex ($\log \beta (\text{Fe}(\text{BHT})_2^-) = 25.3$), which is comparable to that of uranyl ($\log \beta (\text{UO}_2(\text{BHT})_2^{2-} = 26.7$) determined in our study. Therefore, iron cations would compete with uranyl for the adsorption to the H_2BHT -based polymer. However, according to Pan et al. (Ind. Eng. Chem. Res. 55, 4313–4320 (2016)) in contrast to the vanadium species, iron complexes could be efficiently hydrolyzed and stripped from the polyamidoxime fibers under acidic pH conditions (up to 95% iron(III) can be eluted using 1M HCl vs. 2% vanadium(V)), suggesting that the adsorbed iron species would not strongly affect the recyclability of H_2BHT -functionalized adsorbent. The following discussion has been added to the text [page 6 (bottom)]:

It is worth noting that in addition to vanadium, iron ions (Fe^{3+}) form strong complexes with imidodioxime, exhibiting high stability constants, e.g. $\log \beta (\text{Fe}(\text{HIDO})_2^-) = 36.0$, which is 8.5 orders of magnitude higher than $\log \beta$ for the corresponding $\text{UO}_2(\text{HIDO})_2^{2-}$ complex (Table 1). A comparison of formation constants for the 1:2 complexes with bis-(hydroxylamino)-1,3,5-triazine functionality ($\log \beta (\text{Fe}(\text{BHT})_2^-) = 25.3^{29}$ and $\log \beta (\text{UO}_2(\text{BHT})_2^{2-} = 26.7$) revealed comparable binding affinity of H_2BHT toward Fe^{3+} and UO_2^{2+} , suggesting possible competition of iron ions with uranyl for adsorption to the H_2BHT -functionalized sorbent material. Nevertheless, it is reasonable to assume that potentially adsorbed iron species would not strongly affect the recyclability of H_2BHT -based polymer, since iron complexes could be efficiently hydrolyzed and stripped from the polyamidoxime fibers under acidic pH conditions without damaging the adsorbent.

REVIEWERS' COMMENTS:

Reviewer #1 (Remarks to the Author):

This manuscript now looks perfect, I appreciate the amendments the authors conducted.

Reviewer #3 (Remarks to the Author):

I am satisfied with responses of authors and recommend the publication in the current form

Title: Siderophore-inspired chelator hijacks uranium from aqueous medium

Reviewer #1:

General Comments:

This manuscript now looks perfect, I appreciate the amendments the authors conducted.

Response to General Comments: We greatly appreciate the very positive evaluation of our work by the reviewer.

Reviewer #3:

General Comments:

I am satisfied with responses of authors and recommend the publication in the current form.

Response to General Comments: We greatly appreciate the reviewer's positive evaluation of our work.